# Context-Aware Deep Reinforcement Learning for Autonomous Robotic Navigation in Unknown Area

**Jingsong Liang**[*]   **Zhichen Wang**[*]   **Yuhong Cao**[†*]   **Jimmy Chiun**
**Mengqi Zhang**   **Guillaume Sartoretti**

**National University of Singapore**
`{jingsongliang, zhichenwang, caoyuhong, jimmy.chiun}@u.nus.edu`
`{mpezmq, guillaume.sartoretti}@nus.edu.sg`

**Abstract:**

Mapless navigation refers to a challenging task where a mobile robot must rapidly navigate to a predefined destination using its partial knowledge of the environment, which is updated online along the way, instead of a prior map of the environment. Inspired by the recent developments in deep reinforcement learning (DRL), we propose a learning-based framework for mapless navigation, which employs a context-aware policy network to achieve efficient decision-making (i.e., maximize the likelihood of finding the shortest route towards the target destination), especially in complex and large-scale environments. Specifically, our robot learns to form a context of its belief over the entire known area, which it uses to reason about long-term efficiency and sequence show-term movements. Additionally, we propose a graph rarefaction algorithm to enable more efficient decision-making in large-scale applications. We empirically demonstrate that our approach reduces average travel time by up to 61.4% and average planning time by up to 88.2% compared to benchmark planners (D*lite and BIT) on hundreds of test scenarios. We also validate our approach both in high-fidelity Gazebo simulations as well as on hardware, highlighting its promising applicability in the real world without further training/tuning.

**Keywords:** deep reinforcement learning, mapless navigation, context-aware decision-making

## 1 Introduction

Autonomous navigation is an essential capability for mobile robots, and can be broadly divided into local and global planning. Local planning typically focuses on short-term collision avoidance, which provides the robot with reactive kinematic commands to navigate through its nearby surroundings [1, 2]. Global planning, on the other hand, requires the robot to consider the broader environmental information and provide movement decisions at a higher level to determine a long-term route towards the target destination [3, 4]. Although many works have studied *map-based navigation*, where the robot relies on prior information about the environment, in this work, we focus on global planning for *mapless navigation*, where a robot starts navigation to the destination in a completely unknown/unmapped environment. Throughout the task,

Figure 1: **Illustration of navigation through an unknown environment.** The ground vehicle generates a feasible trajectory toward the destination relying on the sensory inputs and destination.

---

[*]These authors contributed equally to this work.
[†]Corresponding authors.

7th Conference on Robot Learning (CoRL 2023), Atlanta, USA.

the robot incrementally constructs/updates its partial belief/map of the environment using onboard sensors, to guide its global path planning. The robot is tasked with reaching the target destination as quickly as possible (in other words, exploring the least amount of the environment).

Although map-based navigation has been relatively well-studied [5, 6, 7], mapless navigation remains an open challenge: it requires the robot to explicitly or implicitly predict the potential path to the target destination based on partial knowledge of the environment. For example, conventional approaches often assume the shortest path lies behind the nearest frontier (i.e., the boundary between traversable and unknown areas) to the destination. Such predictions naturally have a significant impact on navigation efficiency; however, improving the accuracy of these predictions is non-trivial, especially in complex scenarios with plenty of dead-ends and dense obstacles. We believe existing approaches lack *context-awareness* when predicting potential paths. Being context-aware means that the robot is capable of adaptively reason about potential global paths according to its belief over the current environment, instead of following a fixed rule for all scenarios that may be significantly different. It also allows the robot to make non-myopic decisions that benefit long-term efficiency and maximize the likelihood of reaching the destination as fast as possible.

In this paper, we investigate and propose a context-aware DRL framework for mapless navigation. Specifically, our robot reasons about the *context* of its belief over the entire known area using an attention-based neural network. During the navigation task, our robot builds and updates its belief (represented as a graph), which serves as input of our neural network. The policy network models the inter-dependencies between distinct areas in the agent's belief/graph, to finally output the next waypoint for navigation. We further propose a graph rarefaction algorithm to filter out redundant nodes and corresponding edges in this graph for more efficient policy learning in large-scale environments. Compared to existing learning-based approaches [8, 9, 10], our model not only outputs a near-optimal policy to sequence movement decisions towards the target destination, but also endows the agent with time-efficient re-planning abilities even in complex, large-scale environments. To assess the performance and generalizability of our approach, we compare our model to representative state-of-the-art baselines in hundreds of simulated maps at three complexity levels, where we highlight improvements up to 61% for average travel time and 88% for average planning time over these baselines. We also validate the effectiveness of our approach in large-scale Gazebo simulations, and deploy our planner on hardware in a real-world scenario. To the best of our knowledge, we are the first work that proposes a DRL-based approach to mapless navigation [11]. Our full code and trained model is available at https://github.com/marmotlab/Context_Aware_Navigation.

## 2   Related Works

**Search-based approaches:** These approaches are primarily based on discrete grids or a graph with node priority evaluation, such as Dijkstra [12] and A* [5]. Given a prior map, they are able to search for near-optimal paths [6]. Dynamic variants of search-based approaches, e.g., D* [13], LPA* [3], D* lite [4], were proposed to approach dynamic planning in unknown environments. In most cases, these algorithms replan at low frequency, by reusing the planned route from the previous search. However, a robot may have to replan a new feasible route upon realizing that the current one is impractical (e.g., reaching a dead-end) [14]. In this case, such incremental search comes at a high computational cost, especially in complex environments. Furthermore, these algorithms are highly dependent on heuristic function, which results in them being neither cognitive nor robust [15].

**Sampling-based approaches:** More recently, the rapidly-exploring Random Tree (RRT) [16] family, which includes original RRT, RRT* [17], and RRT-Connect [18], has been leveraged for path planning due to its low computation costs. Owing to the drawback of random sampling, the generated paths are typically sub-optimal and unstable [19], and are prone to being stuck in local minima, especially in complex environments. To achieve effective sampling, several improvements have been proposed [20, 21, 22, 23]. These approaches perform well in map-based navigation [7], but typically suffer from long planning time in the absence of prior information [24], mainly due to extensive sampling in both known and unknown areas.

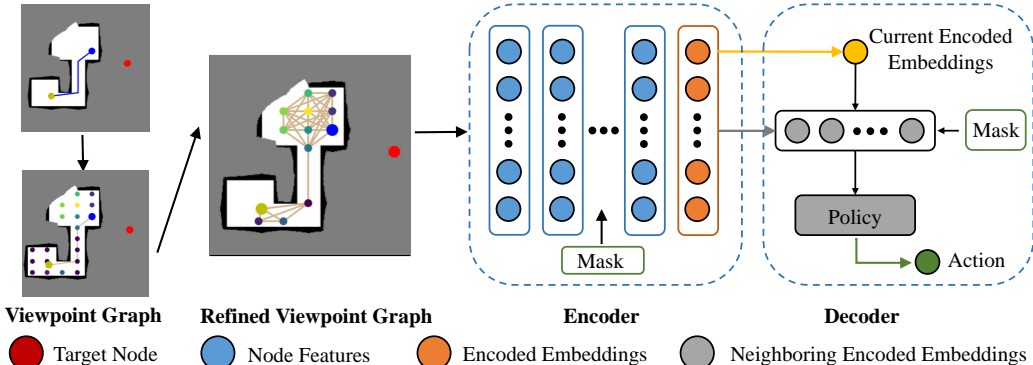

Viewpoint Graph     Refined Viewpoint Graph     Encoder     Decoder

🔴 Target Node    🔵 Node Features    🟠 Encoded Embeddings    ⚪ Neighboring Encoded Embeddings

**Figure 2: Framework of our context-aware DRL policy network.** The initial and refined viewpoint graphs are built from the agent's partial belief and graph rarefaction, respectively. The encoder first incorporates global information from the current partial map and the destination node through self-attention. This embedded information is then used by the decoder to reason about the dependencies between the current node and its neighbors, to finally generate the action policy.

**Learning-based approaches:** Learning-based mapless navigation approaches have aroused interest and demonstrated effectiveness in recent years [2, 9]. They can be broadly subdivided into local and global planning. Most of the previous learning-based works have focused on local planning through end-to-end models [11], including vision-based, i.e., successor-feature-based [8], LiDAR-based [9, 10, 25], or imitation-learning-based approaches [26, 27]. Both [9] and [28] note that a global path planner should usually be used for trust-worthy navigation in unknown environments. Furthermore, these local planners are typically trained and validated in simple environments, raising concerns about their generalizability to more complex cases. To the best of our knowledge, we are the first work that proposes a learning-based global planner for mapless navigation [11].

## 3 Approach

In this section, we consider mapless navigation as a sequential decision-making RL problem and detail our context-aware policy network, as well as our graph rarefaction algorithm to further improve longer-horizon and larger-scale planning.

### 3.1 Mapless navigation as a RL Problem

We formulate the mapless navigation task as a partially observable Markov decision process (POMDP), expressed as a tuple $(\mathcal{S}, \mathcal{A}, \mathcal{T}, \mathcal{R}, \Omega, \mathcal{O}, \gamma)$ with the state space $\mathcal{S}$, the action space $\mathcal{A}$, the state transition function $\mathcal{T}$, the reward function $\mathcal{R}$, the observation space $\Omega$, the observation function given the true state $s'$ of the environment $\mathcal{O}(o_t \in \Omega | s', a)$ and the action state $a$, and the discount factor $\gamma$. To promote efficient navigation towards the target destination, the RL objective is aimed to find an optimal policy $\pi^*$ that maximizes the expected discounted reward $\mathbb{E}_{a_t \sim \pi(\cdot | o_t)} \left[ \sum_{t=1}^{T} \gamma^{t-1} r_t \right]$. The policy $\pi$ can be considered as a mapping function from $o_t$ to the next action $a_t$.

**Observation $\Omega$:** At each decision step $t$, our observation is represented as $o_t = (G_t, S_t)$, which consists of the viewpoint graph $G_t$ and the planning attributes $S_t$. The robot gets the updated observation in the limited sensor range $d_s$ (80 in practice training). The agent first obtains the viewpoint set $V_t$, which is generated uniformly in the known area $\mathcal{D}$. To construct a traversable graph, every node in $V_t$ constructs up to $k$ nearest edges with each other, where the edges in $E_t$ can only connect between collision-free nodes, i.e., nodes that are *line of sight* with each other. The construction of $G_t = (V_t, E_t)$ eliminates the concerns of collision with obstacles. Moreover, the planning attributes $S_t$ provide additional information about the observed environment and the target for each node in $V_t$. Inspired by [29], every node has a direction vector $\vec{v}$ which acts as a signpost for the target, which consists of a unit vector $\hat{v}$ indicating the direction towards the target and the Euclidean distance

$|\vec{v}|$ from the node to the target. $S_t$ further include an indicator $\delta_i$ which records whether the node has been visited before. With the knowledge of the former trajectory, the agent can produce a more informed policy. $S_t$ also includes the utility $u_i$ which is referred to as the number of observable frontiers. These frontiers represent the areas with the potential to the target, which are generated at the boundary of the observed and unknown areas. $S_t$ of each node in $V_t$ are formulated as $\{\vec{v_i}, \delta_i, u_i\}$.

It is worth noting that in large-scale environments, $V_t$ would be populated densely. Meanwhile, the information contained within them is sparse. Therefore, we implement a graph rarefaction algorithm (pseudo code in Appendix A) to prune irrelevant nodes and extract key edges of the viewpoints. Specifically, graph rarefaction first clusters the non-zero utility node set $U$ into multiple groups according to the threshold radius $d_{th}$ (30 in practice). Then, the algorithm uses A* to search for the shortest path $\zeta$, from the robot's current position $p_t$ to each group. After that, the refined node set $V^r \subseteq U$ is constructed by waypoints on $\zeta$, which is either out of $d_{th}$ or out of *line of sight*. The computation complexity is $O(M + NK^d)$, where $M$ is the number of non-zero utility nodes, $N$ the number of nodes chosen to compute the A* path, $K$ the number of edges for each node, and $d$ the number of nodes on the resulting path. After the graph rarefaction process, the refined viewpoint graph $G^r = (V^r, E^r)$ would represent the complete information in the current robot belief. Finally, the $G^r$ along with the corresponding planning states $S^r$ are concatenated as the input of the policy network, i.e., the refined observation $o_s = (G^r, S^r)$.

**Action $\mathcal{A}$:** The collision-free graph extends incrementally along with the update of the agent's observations $o_t$ for every decision step. Our context-aware policy network outputs the stochastic policy $\pi$ for the agent to select the next waypoint among the neighboring nodes. Then, the agent updates its partial map while moving to the next waypoint.

**Reward $\mathcal{R}$:** To promote efficient navigation, the agent receives a reward consisting of three parts. The first part $r_s$ is a constant time step penalty $r_s$ ($-0.5$ in practice). The second part $r_b = d(s_{t-1}) - d(s_t)$ provides continuous feedback on the robot's proximity to the target destination, where $d(s)$ is the distance between the agent current position $p_t$ and the destination location computed by A*. $r_b$ is used to encourage the agent to reach the destination as fast as possible. The last part $r_f$ is a fixed finishing reward, set to 20 while reaching the destination and 0 otherwise. To sum up, the overall reward is $r_t(o_t, a_t) = r_s + c_b \cdot r_b + r_f$, where $c_b$ is a scaling parameter ($c_b = 1/64$ in practice).

### 3.2 Policy Network

Inspired by [30, 31], we design an attention-based neural network (shown in Fig. 2) to sequence efficient movement decisions towards the destination. In our policy network, the encoder embeds the overall information of the current partial map, and the decoder utilizes the learned global features to reason about the dependencies between the current node and its neighbors and finally output an action policy (probability distribution over neighboring nodes).

**Encoder:** The refined observation is first normalized and projected into $d$-dimensional (128 in practice), termed $h^n$. The node features $h^n$ are then passed into multiple attention layers (6 in practice) to aggregate the spatial representation of the current observation. The input of each attention layer consists of a query vector $h^q$ and a key-value vector $h^{k,v}$ of the same dimension. The attention layer updates the query vector with the weighted sum of the value, where the attention weight depends on the similarity between the query and key. In each attention layer, the query $q_i$, key $k_i$ and value $v_i$ are first calculated as $q_i = W^Q h_i^q$, $k_i = W^K h_i^{k,v}$ and $v_i = W^V h_i^{k,v}$ respectively, where $W^Q$, $W^K$, $W^V \in \mathbb{R}^{d \times d}$ are learnable matrices. Next, the similarity between the query $q_i$ and the key $k_j$ is computed with a scaled dot product as $u_{ij} = \frac{q_i^T \cdot k_j}{\sqrt{d}}$. The attention weights $a_{ij}$ are then obtained using a softmax function: $a_{ij} = \frac{e^{u_{ij}}}{\sum_{j=1}^{n} e^{u_{ij}}}$. Finally, the output embedding from an attention layer, denoted as $h_i'$, is calculated as the weighted sum of value vectors as $v_j$: $h_i' = \sum_{j=1}^{n} a_{ij} v_j$. Additionally, an encoder edge mask $M$ is applied to prevent each node from accessing its neighboring features. The output of the encoder, which we term as the *encoded embeddings* $\hat{h}^e$, provides condensed spatial information to the decoder.

**Decoder:** In the decoder, the encoded embeddings of the current node (termed *current encoded embeddings*, $\hat{h}^c$) are first extracted from the encoder as well as its neighboring encoded embeddings (termed *neighboring encoded embeddings*, $\hat{h}^n$). Then the current encoded embeddings and encoded embeddings are fed into an attention layer with $h^q = \hat{h}^c, h^{k,v} = \hat{h}^n$. This layer calculates the output attention weights, representing the relevance of each neighboring node to the current node. These attention weights are concatenated with $h^c$ and projected back to the $d$-dimensional feature together, termed *current decoded embeddings*, $\tilde{h}^c$. The current decoded embeddings incorporate information from neighboring nodes into the representation of the current node. Finally, the current decoded embeddings and the encoded embeddings of neighboring nodes are fed into a pointer layer [32], which is an attention layer that directly uses the normalized attention weights $\theta$ as output.

## 3.3 Training Settings

Inspired by Chen et al. [33], we implement a random dungeon map dataset for training, where there are a total of 800 maps (each map is $1000 \times 1000$ $pixels$) like figure 3. To build the updated collision-free graph, the agent treats the points in the known free area as candidate viewpoints from 1600 points which are uniformly distributed to cover each dungeon map. Our model is trained using the Soft Actor-Critic (SAC) algorithm [34], where the maximum episode step is set to 128, and the size of the replay buffer is set to 10000. The target entropy is set to $0.01 \cdot \log k$, where $k$ represents the number of neighboring nodes. We use the Adam optimizer to optimize policy and critic networks with learning rates of $1 \times 10^{-5}$ and $2 \times 10^{-5}$ respectively. Our model is trained on a workstation with one i9-10980XE CPU and one NVIDIA GeForce RTX 3090 GPU, and the training starts after collecting 2,000 steps in the replay buffer. We utilize Ray [35], a distributed framework for machine learning to parallelize data collection. Training requires around 12 hours to converge.

# 4 Experiments

We set a timeout, i.e., maximum decision steps (128 in practice), to prevent infinite navigation scenarios during testing, and a test is considered a failure if it exceeds this limit. To obtain a complete picture, we report the main performance metrics for mapless navigation, including the average success rate $S(p)$, average travel distance $D(m)$, and average travel time $T(s)$ (including *failed cases* for both latter ones). In particular, $T(s)$ is the sum of each algorithm's step planning time $T_p(s)$ and the robot's resulting motion execution time $T_e(s)$.

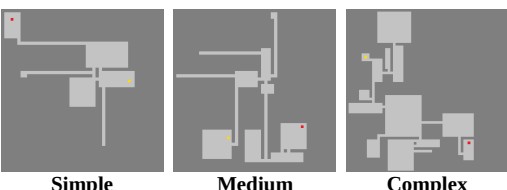

**Simple**   **Medium**   **Complex**

Figure 3: **Examples scenarios with different complexity,** showing the occupied area (grey cells), free space (white cells), start points (yellow block), and target destination (red block).

We first compare our approach with state-of-the-art conventional baselines in numerous dungeon environments (Fig. 3). Then, we compare our model, some variants of our model, and the FAR planner (referred to as "FAR") [24] in a large-scale Gazebo environment (Fig. 5). Lastly, we deploy our trained model on hardware in a real-world scenario (Fig. 6).

## 4.1 Comparisons in dungeon environments

To ensure a fair comparison, we create a random set of testing environments using the random dungeon map generator [33], which were never seen by our trained model. Fig. 3 shows the diverse complexity of testing environments, which can be categorized into three types (50 scenarios each), noted as simple, medium, and complex. We define the quantitative criteria of the scenario complexity as follows: (i) The Euclidean distance between the start point and target destination in each scenario. (ii) The overall number of connecting corridors in all rooms. (iii) The number of intersections that will be encountered along the visually-optimal path. We consider two search-based approaches, LPA* [3], D* lite [4], as well as two sampling-based approaches: RRT [16], and BIT [23].

Table 1: **Comparison results with state-of-the-art baselines in simple, medium, and complex scenarios (50 scenarios for each test set, standard deviation in parentheses).** Environments are randomized $200 \times 200 \ m^2$ dungeons, the LiDAR's scanning range is $30 \ m$, the robot's constant velocity $1.0 \ m/s$, and the graph connectivity parameter $k = 20$ for our model.

| | Criteria | Ours | D* lite | LPA* | RRT | BIT |
|---|---|---|---|---|---|---|
| **S** | $D(m)$ | $224(\pm72)$ | $\mathbf{214}(\pm83)$ | $293(\pm116)$ | $398(\pm208)$ | $275(\pm166)$ |
| | $T(s)$ | $\mathbf{285}(\pm90)$ | $383(\pm152)$ | $414(\pm206)$ | $662(\pm381)$ | $404(\pm237)$ |
| | $T_p(s)$ | $\mathbf{0.20}(\pm0.08)$ | $1.24(\pm0.56)$ | $0.74(\pm0.47)$ | $0.67(\pm0.54)$ | $0.55(\pm0.46)$ |
| | $S(p)$ | $\mathbf{100\%}$ | $\mathbf{100\%}$ | $\mathbf{100\%}$ | $94\%$ | $98\%$ |
| **M** | $D(m)$ | $259(\pm83)$ | $\mathbf{237}(\pm33)$ | $383(\pm130)$ | $497(\pm222)$ | $360(\pm152)$ |
| | $T(s)$ | $\mathbf{329}(\pm105)$ | $484(\pm87)$ | $594(\pm298)$ | $852(\pm565)$ | $563(\pm285)$ |
| | $T_p(s)$ | $\mathbf{0.18}(\pm0.12)$ | $1.52(\pm0.73)$ | $0.79(\pm0.55)$ | $0.68(\pm0.48)$ | $0.53(\pm0.44)$ |
| | $S(p)$ | $\mathbf{100\%}$ | $98\%$ | $94\%$ | $92\%$ | $98\%$ |
| **C** | $D(m)$ | $375(\pm119)$ | $\mathbf{349}(\pm94)$ | $466(\pm163)$ | $544(\pm222)$ | $493(\pm215)$ |
| | $T(s)$ | $\mathbf{477}(\pm151)$ | $680(\pm221)$ | $754(\pm265)$ | $1063(\pm782)$ | $806(\pm407)$ |
| | $T_p(s)$ | $\mathbf{0.22}(\pm0.13)$ | $1.66(\pm0.95)$ | $0.83(\pm0.60)$ | $0.62(\pm0.42)$ | $0.55(\pm0.45)$ |
| | $S(p)$ | $\mathbf{98\%}$ | $94\%$ | $88\%$ | $86\%$ | $90\%$ |

| **Ours** | **D* lite** | **LPA*** | **RRT (partial)** | **BIT (partial)** |

Figure 4: **Trajectory visualization in a representative complex scenario.** The blue line is the robot's trajectory starting at the yellow dot and ending at the red dot. The green lines in RRT and BIT are sampling trees in known areas, which are not included in $D(m)$.

Evaluation results are reported in Table 1. For search-based approaches, our model outperforms D* lite in terms of $S(p)$, $T(s)$, and $T_p(s)$. Despite D* lite showing a lower $D(m)$, our model still surpasses it in terms of $T(s)$ by 15-32%, primarily due to the significantly shorter $T_p(s)$, which is reduced by 83-88%. D* lite is known for finding near-optimal paths, by searching and assessing node priorities incrementally. However, our results illustrate that D* lite results in high computation times (the worst $T_p(s)$ among all approaches), which prevents its use for real-time planning. Moreover, our model outperforms LPA* in three criteria by a substantial margin of approximately 23%, 45%, and 73%. For sampling-based approaches, our model surpasses RRT and BIT in all criteria. Our model finishes the task more than 30-41% faster than BIT, and more than 56-61% faster than RRT. Additionally, our model demonstrates superior performance in terms of $D(m)$, with improvements ranging from 18-28% to BIT and 31-49% to RRT. There, our results illustrate that both RRT and BIT are prone to generate inefficient trajectories, which consequently increases $D(m)$ and $T(s)$. Furthermore, we also conduct ablation experiments (reported in Appendix B) regarding the graph rarefaction algorithm and design of the encoder.

Fig. 4 shows an example where our planner generates a more efficient trajectory, while other baselines suffer from the misleading placement of the target destination. There, our model can be seen to exhibit more interest in unknown areas, with a higher drive toward the destination. We believe that this strategy substantially reduces the likelihood of aimless exploration and consistently aids in generating superior navigation trajectories.

## 4.2 Comparisons in Gazebo environment

We evaluate our planner and some variants (see Table 2) in a highly convoluted environment characterized by narrow corridors and various obstacles. The robot is required to navigate through a series

Table 2: **Evaluations of our model and baselines in large-scale, complex Gazebo environment** (130 $m$ × 100 $m$). We conduct these experiments to evaluate the performance of our model and FAR [24] (An efficient planning framework capable of handling path planning in unknown environments). The robot's constant velocity is 2.0 $m/s$, the LiDAR's scanning range is 30 $m$, and the $D(m)$ and $T(s)$ are the overall values after traveling to 7 successive goals. Our models were trained with $k = 20$, and used as is (no extra training) for the $k = 5, 10$ tests.

|  | **Ours** | **FAR** | **Ours (no GR)** | **Ours ($k = 10$)** | **Ours ($k = 5$)** |
|---|---|---|---|---|---|
| $D(m)$ | **1367.2** | 2060.1 | 1483.5 | 1849.2 | 2103.6 |
| $T(s)$ | **850.7** | 1359.4 | 1131.5 | 1112.1 | 1382.0 |
| $T_p(s)$ | 0.83(±0.37) | 2.42(±0.35) | 1.96(±0.29) | 0.38(±0.11) | **0.36(±0.12)** |

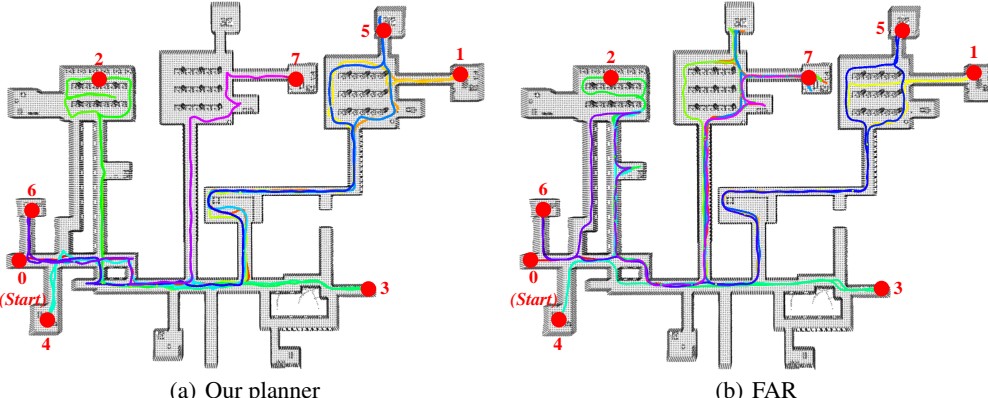

(a) Our planner        (b) FAR

Figure 5: **Trajectories comparison of two planners in large-scale indoor Gazebo environment.**

of predefined target points successfully, similar to the setting presented in FAR [24]. To ensure fairness and consistency in the evaluations, we reset the planners after reaching each target point, which ensures each navigation task begins in a fully unknown environment.

As shown in Fig. 5(a) and 5(b), our model is capable of doing more informed exploration and generating efficient trajectories towards the target destinations. Our experimental results (see Table 2) demonstrate that the significant reduction achieved by our model in terms of $D(m)$ and $T(s)$, by up to 692.9$m$ and 508.7$s$, when compared to FAR. It provides evidence that our model excels in generating more optimal paths than FAR in unknown environments. Additionally, our model outperforms FAR in terms of $T_p(s)$, needing only 0.83$s$ per planning step.

In addition, we conduct further comparisons among several variants of our model, where we vary the number of neighboring nodes $k$ (20, 10, and 5 respectively). Our experimental results indicate that $k = 10$ surpasses FAR in all criteria, without additional training. Furthermore, $T_p(s)$ with $k = 5$ is less than that with $k = 10$ by only 0.02$s$, while the $D(m)$ of $k = 5$ is larger than $k = 10$ up to 254.4$m$, indicating much less efficient trajectories with $k = 5$. Thus, it is not wise to set $k$ as a smaller value due to too sparse connectivity between nodes. To evaluate the significance of graph rarefaction, we test a variant of our model without sparsification, which results in more than 136% $T_p(s)$ and 8% $D(m)$ compared to $k = 20$. Therefore, we believe that graph rarefaction significantly benefits our model.

### 4.3 Hardware Validation in a Real-World Scenario

Our ground vehicle utilizes a Leishen C16 LiDAR for localization and mapping (shown in Fig. 6). The real-world scenario is a 60 × 15 $m^2$ laboratory with randomly placed obstacles (e.g., chairs, boxes, and camera tripod). The vehicle should travel through a series of predefined points by following the waypoint published by our model. After receiving a waypoint, the local planner [36] generates real-time and feasible motion commands for the ground vehicle. The experimental trajec-

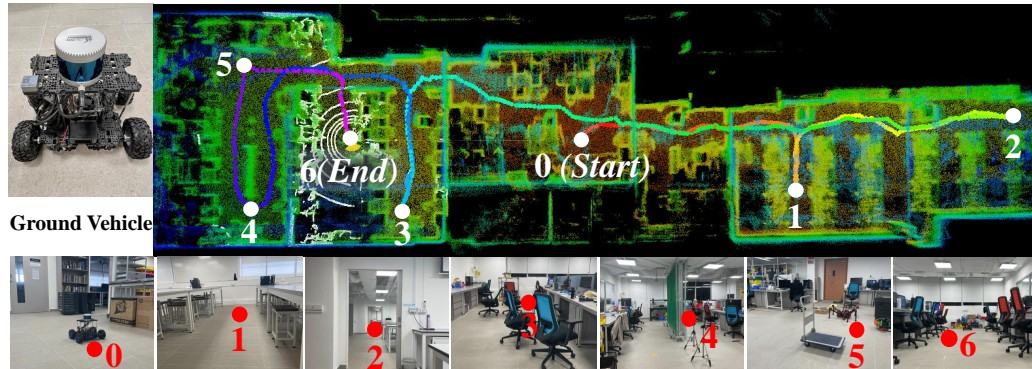

Figure 6: **Validations in real-world environment.** The ground vehicle, provided with partially observed point-cloud data, starts at $point$ 0 and subsequently traverses a series of consecutive points.

tory depicts a high-quality solution, which validates our model's effectiveness and shows promising applicability for real-world environments.

## 5   Limitations

The limitations of our method mainly revolve around adaptive sampling and smooth motion control:

- We currently sample the observations uniformly, which may not precisely capture the spatial representation of the environment and leads to sub-optimal navigation strategies. To tackle this, future work will develop an online adaptive sampling strategy.
- Our model currently plans the next waypoint under the assumption that the robot is omni-directional (i.e., has no motion constraints), which may make reaching this waypoint difficult in practice. The incorporation of a local motion planner (and its use to inform/train the policy) may facilitate navigation in unknown environments with dynamic obstacles.

## 6   Conclusion

In this paper, we propose a context-aware DRL framework for mapless navigation that allows a robot to build a *context* of its entire partial belief over the environment, to infer the shortest route towards a target destination. Our model achieves high-quality decision-making, especially in complex and large-scale environments, where it allows the robot to sequence short-term movement decisions informed by global information about known areas. We also propose a graph rarefaction algorithm to filter out redundant nodes and corresponding edges in the graph input of our neural network, towards deployment in large-scale environments. We empirically demonstrate that our model outperforms state-of-the-art baselines in terms of average travel time and average planning time, with powerful generalizability to complex unknown environments never seen during training. Finally, we validate our approach in high-fidelity Gazebo simulations as well as on hardware, revealing promises for robotic deployments in the real world without further tuning.

Future work will first focus on the construction of a local planner with more considerations on kinematic/dynamics constraints. Then, we will extend our framework to multi-agent mapless navigation, where robots need to reason about each other and plan efficient paths cooperatively, by leveraging synergies and avoiding redundant work.

**Acknowledgments**

This work was supported by Temasek Laboratories (TL@NUS) under grant TL/FS/2022/01.

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

# Appendix

## A    Graph Rarefaction

---
**Algorithm 1:** Graph Rarefaction Algorithm

---
**Input:** non-zero utility node set $U$, map $\mathcal{M}$, robot position $p_t$, threshold radius $d_{th}$

Initialize refined node set $V^r \leftarrow U$, covered node set $\overline{U} \leftarrow \emptyset$;

**for** $v \in U$ **do**

    **if** $v \in \overline{U}$ **then** continue;

    Find nearby node set $N$ in $d_{th}$;

    **for** $v' \in N$ **do**

        | **if** $line(v, v')$ *is collision free* **then** $\bar{U} \leftarrow v'$;

    **end**

    Find path $\zeta$ from $p_t$ to $v$, set ref node $v_{ref} = v$;

    **for** $i \in |\zeta|$ **do**

        | **if** $line(v_{ref}, \zeta_i)$ *is not collision free or* $L\left(v_{ref}, \zeta_i\right) \geq d_{th}$ **then**

        |     | $v_{ref} = \zeta_{i-1}, V^r \leftarrow v_{ref}$

    **end**

**end**

**Output:** collision free edge set $E^r$ based on $V^r$ and $\mathcal{M}$.

---

## B    Ablation experiments

We carried out ablation experiments on our model to evaluate the impact of its key parameters. We first introduced the various ablation cases, which are our model with the original setting (termed **Ours**), our model with original viewpoint graph (i.e., without graph rarefaction, termed **Ours (no GR)**), and our model with two encoder layers (termed **Ours (Encoder-2)**). All the cases are evaluated in 2D environments (same test set as in the paper) and reported in Table 3.

Table 3: **Ablation experiments on our model in simple (S), medium (M), and complex (C) scenarios (50 maps per test set).** The comparison metrics are average travel distance $D(m)$, motion execution time $T(s)$, step planning time $T_p(s)$, and test success rate per scenario $S(p)$, respectively. The standard deviation of the comparative metrics is indicated by the parentheses.

|   | Criteria | Ours | Ours (no GR) | Ours (Encoder-2) |
|---|---|---|---|---|
| **S** | $D(m)$ | **224**($\pm$72) | 245($\pm$55) | 382($\pm$198) |
|  | $T(s)$ | **285**($\pm$90) | 313($\pm$101) | 489($\pm$253) |
|  | $T_p(s)$ | 0.20($\pm$0.08) | 0.34($\pm$0.06) | **0.18**($\pm$0.10) |
|  | $S(p)$ | **100%** | **100%** | 98% |
| **M** | $D(m)$ | **259**($\pm$83) | 276($\pm$104) | 443($\pm$233) |
|  | $T(s)$ | **329**($\pm$105) | 452($\pm$126) | 561($\pm$294) |
|  | $T_p(s)$ | 0.18($\pm$0.12) | 0.31($\pm$0.09) | **0.13**($\pm$0.03) |
|  | $S(p)$ | **100%** | **100%** | 88% |
| **C** | $D(m)$ | **375**($\pm$119) | 410($\pm$127) | 611($\pm$240) |
|  | $T(s)$ | **477**($\pm$151) | 573($\pm$235) | 774($\pm$305) |
|  | $T_p(s)$ | 0.22($\pm$0.13) | 0.38($\pm$0.15) | **0.16**($\pm$0.04) |
|  | $S(p)$ | **98%** | 96% | 84% |

As shown in Table 3, the model **Ours (no GR)** degrades slightly in terms of $D(m)$, $T(s)$, and $S(p)$ compared to our original model. However, we note that our original model achieves a computational efficiency that is twice as fast as the step planning time $T_p(s)$, which indicates the significance of the graph rarefaction algorithm in our approach. Moreover, we find that the model **Ours (Encoder-2)** exhibits a large drop in performance, particularly in the medium and complex scenarios. This result indicates that an encoder with more layers assists the robot in avoiding myopic decisions, particularly on long-term mapless navigation tasks.

