# OpenReview forum: "Context-Aware Deep Reinforcement Learning for Autonomous Robotic Navigation in Unknown Area"
_robot-learning.org/CoRL/2023/Conference — CoRL 2023 Poster_

### Official Review · Reviewer_cs7G · 2023-06-30

**Confidence:** 4
**Originality:** Good
**Technical Quality:** Good
**Clarity Of Presentation:** Good
**Impact:** 3

**Recommendation:**

Weak Accept: I recommend accepting the paper, but will not argue for my recommendation if the majority of other reviewers have a different opinion.

**Review:**

Strengths:

+ The paper is in general well written and seems to be technically correct
+ It performs better than traditional planners (e.g., RRT) in terms of runtime and traveled distance

Weakness:
-  The comparative experiments are not convincing. Comparisons with respect to mission success rates are missing, which is probably a much more important metric than runtimes. In fact, it is not surprising that a learned policy will be faster and more efficient than a traditional planning policy (e.g., RRT). This is critical as the paper claims that existing planners lack of stability and optimality but there is no evidence that the proposed method addresses these shortcomings.
- Comparisons against other learning-based approaches is missing. The authors claim that existing method generalize poorly to new environments. This is in general an issue in related works but there are no comparisons showing improvement over related works.
- The novelty of the paper compared to related works is vague.


**Quality Of The Limitations Section:**

Additional details required

**Questions For Rebuttal:**

1) Why is it called "context-aware" DRL? Is it because of the graph rarefaction algorithm? The authors need to explicitly explain what "context" means, how it is utilized, and what are its benefits.

2) What does mapless navigation mean? Does it just mean that the environment is unknown or it additionally implies that planning is performed in the unknown environment without actually learning the map of the environment (e.g., an occupancy grid map)?

3) Although the literature review is complete, it is unclear if the proposed method really address the existing challenges. For example, it is claimed that sampling based methods are suboptimal, unstable, and they may get stuck in local minima  (although asymptotically they will find a feasible solution). Although this is true, several methods have been proposed to overcome such challenges e.g., by developing intelligent sampling strategies [A], [B] or hallucinating the unknown environment [C]. Such methods are not discussed. At the same time, the proposed does not necessarily address the above challenges (even though the the empirical performance is satisfactory). The authors do not need to cite these specific papers but they need to explain better how this work fits in the existing planning literature

[A] "Perception-based temporal logic planning in uncertain semantic maps." IEEE Transactions on Robotics 38.4 (2022): 2536-2556.

[B] "Guided Sampling-Based Motion Planning with Dynamics in Unknown Environments." arXiv preprint arXiv:2306.09229 (2023).

[C] "Map-predictive motion planning in unknown environments." 2020 IEEE International Conference on Robotics and Automation (ICRA). IEEE, 2020.

4) Similarly, the authors claim that existing learning-based planners cannot be applied to large and unseen environments. This is not entirely correct. Most existing learning-based planners do not have any provable scalability/generalization guarantees but they have quite impressive empirical performance. See e.g. [D]. As before, the authors do not need to cite this specific paper. However, a more fair presentation of existing works is needed.

[D] "Socially aware motion planning with deep reinforcement learning." 2017 IEEE/RSJ International Conference on Intelligent Robots and Systems (IROS). IEEE, 2017.

5) In Section 3.1., is the POMDP state and action space discrete? Does every time the robot knows its current state, i.e., the state s it belongs to? In general, it's a bit unclear how the POMDP models an unknown environment and what exactly is unknown in the environment. E.g., is the state transition function T known?

6) What assumptions are made about the observation function? Is it assumed to be a perfect observation/sensor model? I.e., is its output always correct? How can it implemented in practice?

7) In Section 3.2, what exactly is the input to the neural network policy? Is it the graph Gt and the current robot state?

8) What is the target entropy in Section 3.3? How is it used?

9) In the comparative experiments, the traveled distance and travel time are used as evaluation metrics. The authors should mission success rates as well. This is a much more important metric than runtimes and traveled distance. In fact, it is not surprising that a learned policy will be faster and more efficient than a traditional planning policy (e.g., RRT). Also, are the reported results based on the average performance of multiple runs or a single run? If the former, in the reported travel times/distances, are all runs considered, or only the ones where the learned policy succeeds?

10) In figure 4, why do the blue paths for D* and LPA* initially move quite far away from the target? That result looks somewhat incorrect given that these methods treat the unknown space as free space. Also, how are the baselines executed? Is replanning triggered at every single iteration or only at time steps where the current paths intersect with unexpected obstacles? This is critical to fairly evaluate the runtimes of these methods; note that replanning at every time step may not be necessary.

11) How does the proposed algorithm perform against other learning-based planners? Does it outperform them in terms of scalability and generalization to unseen environments?

**Robotics Focus:**

Sufficient demonstration on hardware

**Summary Of Paper:**

This paper proposes a new DRL framework for navigating unknown environments. The environment is modeled as a graph that is continuously learned. To mitigate the large graph space, a pruning method is used that improves path planning efficiency. This graph is used by a neural network policy, learned using DRL, to learn planning policies. The developed  algorithms outperforms related motion planners in terms of runtime and traveled distance.

**Summary Of Recommendation:**

Overall, the paper proposes an interesting approach to solve navigation problems in unknown environments. However, the comparative experiments are not convincing. Additional experiments are needed to demonstrate the benefits of this methods over existing ones.

---

### Official Review · Reviewer_eMFY · 2023-07-11

**Confidence:** 3
**Originality:** Good
**Technical Quality:** Good
**Clarity Of Presentation:** Very Good
**Impact:** 4

**Recommendation:**

Weak Accept: I recommend accepting the paper, but will not argue for my recommendation if the majority of other reviewers have a different opinion.

**Review:**

Strengths

-  A well-written paper that is a pleasure to read.
-  The problem is well-motivated and of relevance to the CoRL community.
-  Extensive evaluation against state-of-the-art methods in synthetic datasets.

Weaknesses

-  In the discussion of the result from the hardware experiment, it is stated that the robot ends up following a near-optimal solution.  I would justify why this statement holds true: since no comparison with other methods is included as part of the experiment, it appears hard to infer how near-optimal the solution is, and whether the solution's quality is due to the used local planner or due to the context-aware part of the proposed method.  Is it the local planner that primarily enables the resulting trajectory or the context-aware part of the proposed method?

-  Since the proposed method is based on deep learning, I would say that one can find corner cases where the method fails.  Although the current synthetic experiments suggest good generalizability of the method, could a discussion be included in the paper discussing corner cases and potential remedies?

-  I would make explicit early in the paper what the "context" is and how it is integrated into the algorithm.

**Quality Of The Limitations Section:**

Limitations are not well addressed

**Questions For Rebuttal:**

Please see my comments in the weaknesses section above.  Also, for the hardware experiment, what is the training set?  Is it the same as for the 2D-map synthetic experiments?

**Robotics Focus:**

Sufficient demonstration on hardware

**Summary Of Paper:**

The paper proposes a deep-reinforcement-learning method for guiding a mobile robot to reach a predefined target in an unknown environment.  The proposed method is evaluated against state-of-the-art methods in extensive simulator experiments.  The proposed method is observed to significantly reduce planning and travel time in the simulator experiments.  Also, a hardware demonstration is presented, without comparison with state-of-the-art methods.

**Summary Of Recommendation:**

The paper focuses on a fundamental robotics and learning problem, proposing a method that appears promising for real-world deployment. In my opinion, the current version of the paper would benefit from (i) discussing corner cases, (ii) clarifying how the proposed method can be trained for real-world deployment, and (iii) providing supporting evidence of its superiority during real-world deployment.

UPDATE: I keep my original recommendation (Weak Accept) upon accounting for the generalization weaknesses of deep learning methods.

---

### Official Review · Reviewer_CUZW · 2023-07-20

**Confidence:** 4
**Originality:** Fair
**Technical Quality:** Fair
**Clarity Of Presentation:** Fair
**Impact:** 2

**Recommendation:**

Weak Reject: I recommend rejecting the paper, but will not argue for my recommendation if the majority of other reviewers have a different opinion.

**Review:**

Strengths

Strong Apparent Performance Characteristics
The comparative results between the proposed DRL method and classical methods in the literature displays the utility of learning-based methods in providing reasonable solutions to what would otherwise be complex, time consuming queries.  The model is shown to be
substantially faster than implementations of classic and state-of-the-art heuristic methods (though that comparison would benefit from a more detailed account of software used, so as to confirm that the performance is not limited by the implementation).

Descriptive Figures
The figures provided are useful for clarification and reader understanding of the evaluated environments, as well as the proposed architecture (Fig 2).

Multi-Environment Evaluation
The authors test their method across three environment types/fidelities, including a real-world experiment


Weaknesses

Weak Taxonomic Overview and Embedding in the Literature
In their introduction and discussion of background/related works to that which is proposed here, the authors offer a high-level taxonomy
of approaches to robotic path-planning.  While the trichotomy of search-based, sampling-based, and learning-based approaches described is a valid breakdown of path-planning methodologies, the arguments offered for the shortcomings of previous works in each set of high-level approaches are limited in their persuasiveness.  For example, the array of varying optimal and near-optimal shortest path
approaches are dismissed due to replanning "always" incurring high computation costs, as well as a lack of robustness and cognitivity
from methods which leverage a heuristic function.  The dismissal of traditional sampling-based methods is equally limited, citing limitations of sampling-based approaches that are by no-means universal (oversampling in free and unknown areas).  Further, it is stated that random-sampling in general yields instability and suboptimality in pathing, despite already citing asymptotically-optimal sampling approaches, and ironically citing that claim itself from the LQR-RRT* paper; the body of which is a clear contradiction to this claim.

Unclear Preprocessing Complexity and Utility
In transforming the current map into the agent observation, the authors propose a graph simplification algorithm with aim to avoid
blowup in space-complexity as the scale of the environment is increased (as the viewpoint graph is otherwise a dense 2D-grid).  It
would be helpful to include comparisons between the proposed algorithm and the corpus of existing work in the literature of graph simplification for path-planning.  Furthermore, it is unclear from the paper the computational complexity/overhead incurred by this algorithm at each planning step.  The justification for applying learning to the problem domain is weakened when the preprocessing step to that learned system is itself nontrivial.  The reader would further benefit seeing evaluation of baseline planning methods operating directly over this refined viewpoint graph, rather than a general map space.  (If it is the case that the viewpoint graph is used in the evaluations of baseline search-based algorithms, it is not made clear in the paper as it stands).

Potential for Overfitting to Training Data
When leveraging a learning-based approach to decision-making problems with partial state visibility, one must be deliberate in their efforts to acknowledge the potential for the agent to learn not only the desired task, but also non-generalizable patterns from the training
data/environments.  Leveraging exclusively generated dungeon maps for training, and comparing the agent's path-planning distance to
generalized best-first search is unfair in the general case.  Through training, the agent can learn patterns present in the dungeon-map
generation algorithm, and use this information to improve its  via encoded prior knowledge.  It would benefit the work to see comparisons
among a wider array of simple environment topologies.

Limited Real-World/High-Fidelity Evaluation
While the inclusion of evaluation in a realistic robot simulator and the real-world is beneficial to the paper, the breadth and depth of
study in these environments is too limited to be sufficiently convincing to the reader.  The Gazebo environment simulation would
serve as an excellent grounds for comparison of the traditional algorithms tested in the simple environment, yet only FAR is evaluated
against.  Reviewing the paper which proposed the FAR Planner, the paper would benefit from at least citing the evidence of FAR's
outperformance of the other baselines (though a full study with more baseline algorithms would be even stronger).  Furthermore, the
real-world results generated by the algorithm are presented without comparison, leaving the reader to evaluate the real-world efficacy of the proposed method purely qualitatively.

**Quality Of The Limitations Section:**

Limitations are not well addressed

**Questions For Rebuttal:**

- Could you elaborate further on the embedding of this work into the established literature?  This would include more related work describing learning-based approaches to this task, as well as elaboration/clarification of the heuristic method taxonomy.

- What is the computational complexity of the proposed graph rarification algorithm?  How does it compare to the complexity of some of the full baseline algorithms?

- Can you include more specifics regarding the comparative implementations of the baseline algorithms (D*, RRT, etc.)?

**Robotics Focus:**

Sufficient demonstration on hardware

**Summary Of Paper:**

In this work, the authors describe and implement a method for leveraging deep reinforcement-learning (DRL) for efficient global path
planning in environments lacking a prior map (mapless navigation). The proposed method is touted for its improved performance over
classical path-planning methods, in terms of both required computation time and efficiency (shortness of distance) of planned paths.  The
system is trained on simple, randomly-generated 2D dungeon-map environments, and is evaluated over new 2D maps, a Gazebo simulation environment, and a real-world office-like environment.

**Summary Of Recommendation:**

In summary, this work presents a reasonable and well-motivated algorithm for path-planning using deep reinforcement learning, but lacks sufficient rigor in embedding itself in the prior literature. It furthermore is limited in the scope of its experimental findings/analysis, and in its interrogation of/motivation for replacing classical baseline algorithms with a learning-based approach.

Beneath these shortcomings I believe there could be a valuable contribution to the robotics literature.  Additional focus and clarification lent to these points would do well to form this paper into a much stronger candidate for submission.

Post-rebuttal comments: I would also like to note that the above ratings accurately reflect this reviewer's views of the paper upon reading the authors' rebuttal responses.

---

### Official Review · Reviewer_qPjV · 2023-07-23

**Confidence:** 4
**Originality:** Good
**Technical Quality:** Very Good
**Clarity Of Presentation:** Very Good
**Impact:** 3

**Recommendation:**

Weak Accept: I recommend accepting the paper, but will not argue for my recommendation if the majority of other reviewers have a different opinion.

**Review:**

The proposed approach is an effective method for mapless navigation in unknown environments. The results demonstrate that it is capable of reducing average travel time (i.e., makespan) up to 61.4% and average planning time up to 88.2% when compared to benchmark planners (D*lite and BIT). Additionally, the approach is validated in high-fidelity Gazebo simulation as well as on ground robot, highlighting its promising applicability to real-life environments without further tuning. The encoder embeds the overall information of the current partial map, and the decoder reasons the local information of the current node and its neighboring nodes to finally output the action policy. The policy network is an attention-based neural network that is used to generate efficient path planning towards the target.

The proposed model appears to be effective in generating more efficient paths than the conventional baselines in unknown environments. However, it is unclear if the model is able to handle dynamic obstacles, which may be encountered in real-world scenarios.

The proposed model has been tested and validated in a large-scale Gazebo environment, and the results demonstrate that the model is able to generate more optimal paths than the FAR planner in unknown environments. However, the model may not be able to handle very large-scale environments, and further testing is needed to evaluate its performance in such scenarios.

**Quality Of The Limitations Section:**

Limitations are addressed clearly

**Questions For Rebuttal:**

The authors should add a table of ablation expeiments over their own method to demonstrate the effectiveness of their design.

Missing some of important notations, for example what are the red dots in Figure 2? Also it would be super helpful if the captions explain more major details and ideas of the figures.

In table 2, should be included in caption what is FAR and citation.

In the real world experments shown in the video attachment, the robot never makes a mistake at any intersection even if the goal is invisible. I wonder why this happens? Are there any cases where the robot makes mistakes, and if so I'd recommend adding some of them to the attachment to avoid confusion.

Finally, larger scale map navigation is encouraged to study and add a table of how the performance is like for various tasks of different challengeness.

**Robotics Focus:**

Sufficient demonstration on hardware

**Summary Of Paper:**

This paper proposes a learning-based framework for autonomous robotic navigation in unknown areas. The framework employs a context-aware policy network to extract and utilize spatial information to generate an effective path planning policy. Additionally, a graph rarefaction algorithm is proposed to remove irrelevant or redundant information in the navigation environment. The approach allows the agent to learn an efficient path planning policy, which can re-plan at low latency while receiving new environmental observations. The approach is evaluated on hundreds of test scenarios and compared to benchmark planners (D*lite and BIT). Results show that the approach reduces average travel time (i.e., makespan) up to 61.4% and average planning time up to 88.2%. The approach is also validated in high-fidelity Gazebo simulation as well as on ground robot, highlighting its promising applicability to real-life environments without further tuning.

**Summary Of Recommendation:**

This is a solid technical report of the authors' recent work on incorporating reinforcement learning and graph representation for navigation. Although there are already plenty of works in this field. Both the experiments and paper organizaiton are convincing, although the paper lacks some of the important comparisons with other RL based navigation works and ablation of their own designs. In my opinion this paper is around the borderline.

---

### Decision · Program_Chairs · 2023-08-30

**Decision:**

Accept (Poster)

**Comment:**

The authors  present a novel approach to address the challenge of mapless navigation in unknown environments using deep reinforcement learning. The paper is well-written and clearly motivates the problem, offering a detailed overview of the proposed methodology. The reviewers have provided valuable feedback and raised several critical points that need to be addressed for the paper's improvement. The authors' responses indicate a thorough understanding of the reviewers' comments, and they have proposed revisions that will significantly enhance the clarity, technical quality, and comprehensiveness of the paper.

Strengths:
The paper introduces an intriguing approach to mapless navigation in unknown environments using deep reinforcement learning. The problem is well-motivated, and the paper clearly outlines the benefits of the proposed approach. The technical quality of the work is evident, and the authors demonstrate a strong grasp of the theoretical foundations and practical implementation of their method. The paper is generally well-organized and clearly presented, making it accessible to readers from various backgrounds. The inclusion of both simulation and real-world hardware experiments further strengthens the validity of the proposed approach.

Weaknesses:
The weaknesses highlighted by the reviewers primarily revolve around the need for additional explanations, discussions, and experimental analyses. The lack of comparisons with other learning-based planners, the assumption of perfect localization and mapping, and the absence of corner cases are notable issues that need to be addressed. The concerns raised about the comparative experiments, the necessity of mission success rates, and the handling of search-based algorithms' replanning behavior should also be considered for improvement. Furthermore, clarity on the concept of "context-aware" reasoning and elaboration on the training process for real-world deployment are areas that require further attention.

Reviewer-Specific Comments:
Suggestions regarding embedding the work in the established literature, computational complexity, and more specifics on baseline implementations should be addressed in the revised manuscript.
Feedback about discussing corner cases, clarifying the training process, providing mission success rates, and addressing the absence of comparisons with other learning-based planners should be taken into account.
Additional questions about the POMDP formulation, observation function assumptions, policy network input, target entropy, and graph movement in Figure 4 need to be thoroughly addressed.

The authors' responses indicate a clear understanding of the reviewers' feedback and their commitment to improving the paper. The planned inclusion of more comprehensive discussions on related work, addressing the concept of "context-aware" reasoning, providing clarifications on the training process for real-world deployment, and incorporating a detailed explanation of the POMDP formulation are positive steps towards enhancing the manuscript's technical quality and clarity. The authors' efforts to address corner cases, provide mission success rates, and clarify replanning behavior for search-based algorithms are also commendable.